# Structural basis of specific H2A K13/K15 ubiquitination by RNF168

Velten Horn[1,5,6], Michael Uckelmann[2,3,6], Heyi Zhang[1,5], Jelmer Eerland[1], Ivette Aarsman[2], Ulric B. le Paige[1,5], Chen Davidovich [3,4], Titia K. Sixma[2] & Hugo van Ingen[1,5]

Ubiquitination of chromatin by modification of histone H2A is a critical step in both regulation of DNA repair and regulation of cell fate. These very different outcomes depend on the selective modification of distinct lysine residues in H2A, each by a specific E3 ligase. While polycomb PRC1 complexes modify K119, resulting in gene silencing, the E3 ligase RNF168 modifies K13/15, which is a key event in the response to DNA double-strand breaks. The molecular origin of ubiquitination site specificity by these related E3 enzymes is one of the open questions in the field. Using a combination of NMR spectroscopy, crosslinking mass-spectrometry, mutagenesis and data-driven modelling, here we show that RNF168 binds the acidic patch on the nucleosome surface, directing the E2 to the target lysine. The structural model highlights the role of E3 and nucleosome in promoting ubiquitination and provides a basis for understanding and engineering of chromatin ubiquitination specificity.

[1] Department of Macromolecular Biochemistry, Leiden Institute of Chemistry, Leiden University, P.O. Box 95022300 RA Leiden, the Netherlands. [2] Division of Biochemistry and Oncode Institute, Netherlands Cancer Institute, P.O. Box 902031006 BE Amsterdam, the Netherlands. [3] Department of Biochemistry and Molecular Biology, Biomedicine Discovery Institute, Faculty of Medicine, Nursing and Health Sciences, Monash University, 23 Innovation Walk, Clayton, VIC 3800, Australia. [4] EMBL-Australia and the ARC Centre of Excellence in Advanced Molecular Imaging, Clayton, VIC 3800, Australia. [5] Present address: NMR Group, Bijvoet Center for Biomolecular Research, Utrecht University, Padualaan 8, 3584 CH Utrecht, the Netherlands. [6] These authors contributed equally: Velten Horn, Michael Uckelmann. Correspondence and requests for materials should be addressed to T.K.S. (email: t.sixma@nki.nl) or to H.v.I. (email: h.vaningen@uu.nl)

The covalent attachment of ubiquitin to target proteins is a key post-translational modification in almost any eukaryotic cellular pathway. The process relies on an E1-E2-E3 conjugation machinery, where the ubiquitin E3 ligases catalyze the discharge of ubiquitin from a charged E2 to a target lysine[1,2]. The E2 discharge is facilitated through conformational selection, where the E3 engages in specific interactions with E2 and ubiquitin. This stabilizes the ubiquitin-charged E2 in a conformation that leaves the active site prone to aminolysis[3–6].

How target lysines are selected and how selective this process is varies. While most E3 ligases likely ubiquitinate target proteins rather unselectively[7,8], some E3 ligases show a remarkable specificity for a certain group of lysines[2,9–11]. Lysine selection here is achieved through specific contacts between the E3 and the target protein. This interaction orients the E2 active site towards the appropriate target lysine[1,2,12]. The E3-substrate interactions that guide lysine selection are likely unique for each E3-substrate pair. An interesting model system to study target lysine selection by the E3 ligase is ubiquitination of nucleosomes, especially of histone H2A where multiple E3 ligases modify distinct sites.

Three separate E3 ligases, RNF168, RING1B/BMI1, and BRCA1/BARD1, each ubiquitinate a unique site on H2A, each with a distinct biological outcome[9–11]. Ubiquitination at lysine 118/119 (K119) by PRC1 polycomb complexes, such as RING1B/BMI1 is important for transcriptional silencing[13] and may play a role in the regulation of DNA double-strand break (DSB) repair[14–16]. Ubiquitination at lysine 13/15 (K15) by RNF168 and at lysines 125/127/129 (K129) by BRCA1/BARD1 both are crucial in the DNA damage response. The K15 mark acts as a binding platform for specific recruitment of 53BP1[17,18], a protein responsible for establishing a DNA-end resection block[19,20]. The BRCA1 mark at K129 leads to recruitment of SMARCAD1, a chromatin remodeler that facilitates extended DNA-end resection[21]. The balance between K15 and K129 ubiquitination is thought to determine the choice between DSB repair by either non-homologous end joining or one of the homology repair pathways[14,21,22].

Initial understanding of how these RING-type E3 ligases can target a nucleosome has come from the spectacular crystal structure of the complex of the E2, the RING domains of RING1B/BMI1 and the nucleosomal core particle. In this structure RING1B/BMI1 interacts with a cluster of acidic residues on the nucleosome surface to orient UbcH5C (Ube2D3), the cognate E2, towards K119[12]. Based on biochemical characterizations a similar mode of interaction was proposed for BRCA1/BARD1, though direct structural evidence is missing in this case[12]. For RNF168, mutagenesis studies have shown that part of the acidic patch is crucial for RNF168 activity but not for direct recruitment[23,24] and that a basic residue on RNF168 (R57) is essential for binding the nucleosome[9,23]. Strikingly, the presence of the nucleosome was observed to promote the rate of ubiquitination reaction, indicating it contributes to E2 activation. While the structure of RING1B/BMI1 in complex with the nucleosome provides valuable insight towards the mechanism of interaction, for RNF168 and BRCA1/BARD1 it is not yet clear how, on a molecular level, the acidic patch is employed to govern unique lysine selectivity, neither how the nucleosome contributes to E2 activation.

Here we provide mechanistic analysis of specific ubiquitination of H2A at K15 by RNF168. Using NMR interaction studies on both histone complexes and nucleosome substrates, we show that the RING domain of RNF168 binds directly to the nucleosome acidic patch. Through ubiquitination assays, we identify multiple critical arginine residues in RNF168. Based on the combination of NMR, mutational analysis and crosslinking mass-spectrometry data, we present a data-driven integrative structure of the E3-substrate complex highlighting how the RING domain RNF168, while bound to the nucleosome, orients the E2 towards the target lysine. The structure suggests that the nucleosome surface contributes to E2 activation by promoting the formation of the closed, activated ubiquitin-charged E2 conformation. Using structure-guided mutagenesis, we identified a single mutation at the periphery of the binding site and distant from the target lysine that is sufficient to specifically interfere with RNF168 ubiquitination, while retaining RING1B activity. Our findings underscore the crucial role of the shape and electrostatic properties of acidic patch binding proteins in determining their precise binding modes and point to the intricate contribution of both E3 ligase as well as substrate in controlling activity and specificity of ubiquitination.

## Results

**RNF168-RING binds the acidic patch using an Arg-rich helix.** RNF168 harbors multiple domains that are critical for its function in the DNA damage response (Fig. 1a). It contains two ubiquitin interacting motifs (Ub-dependent DSB recruitment modules, UDMs): one is necessary for recruitment of RNF168 to the DNA damage site by binding ubiquitinated linker histone H1[25], while the second motif binds to ubiquitinated H2A, the product of RNF168 activity[26–28]. Its monomeric RING domain is required and sufficient to direct ubiquitination by E2 UbcH5c to H2A residues K13/K15[9,23]. To understand the substrate recognition and ubiquitination specificity of RNF168, we set out to identify how its RING domain (RNF168$^{RING}$) interacts with the nucleosome. Despite extensive efforts, structural analysis of the RNF168$^{RING}$-nucleosome interaction through crystallography was unsuccessful. Over recent years NMR spectroscopy has proven to be an attractive approach to map out the binding interfaces involved in nucleosome-protein interactions[29–34]. Since RNF168-UbcH5c can specifically ubiquitinate both H2A-H2B dimers (25 kDa) and nucleosomes (200 kDa), we mapped the interaction surface of the RNF168$^{RING}$ domain on histone dimers using traditional amide-backbone-based NMR and on the nucleosome using methyl-group based NMR for high-molecular weight systems.

Titration of the RNF168$^{RING}$ domain to H2A/H2B dimers with either $^{15}$N-labeled H2A or H2B caused specific changes in the NMR signals of H2A and H2B residues (Fig. 1b and Supplementary Fig. 1). Progressive addition of RING domain also caused increasing levels of precipitation, making it impossible to reach a fully bound state of the dimer. Poor solubility of the RNF168$^{RING}$-H2A-H2B complex was observed for a range of buffer conditions tested. Under the substoichiometric conditions of the titration experiments, the dimer remains predominantly unbound such that the effect of binding is mostly visible through a peak intensity decrease for specific amide groups (Fig. 1c). Resonances with most significant changes include residues L22, Q23, E63, E90, E91, and L92 of H2A and a region around E102 and T116 in H2B (*Dm*. histone residue numbering). These residues cluster in and around the acidic patch on the surface of the H2A/H2B dimer (Fig. 1d). In addition, peak intensity changes and small chemical shift changes were observed for regions of the H2A/H2B surface that are occluded in the context of the nucleosome, for example the DNA-binding region around H2A H30 and the DNA and chaperone binding region around H2B I51 (see Fig. 1c, d and Supplementary Fig. 1). Suspecting that the effects on the latter surfaces are due to unspecific binding of RNF168$^{RING}$, we sought to experimentally verify this. Since specific binding should result in larger chemical shift differences and slower dissociation rates than unspecific binding, we reasoned that these binding modes would result in an

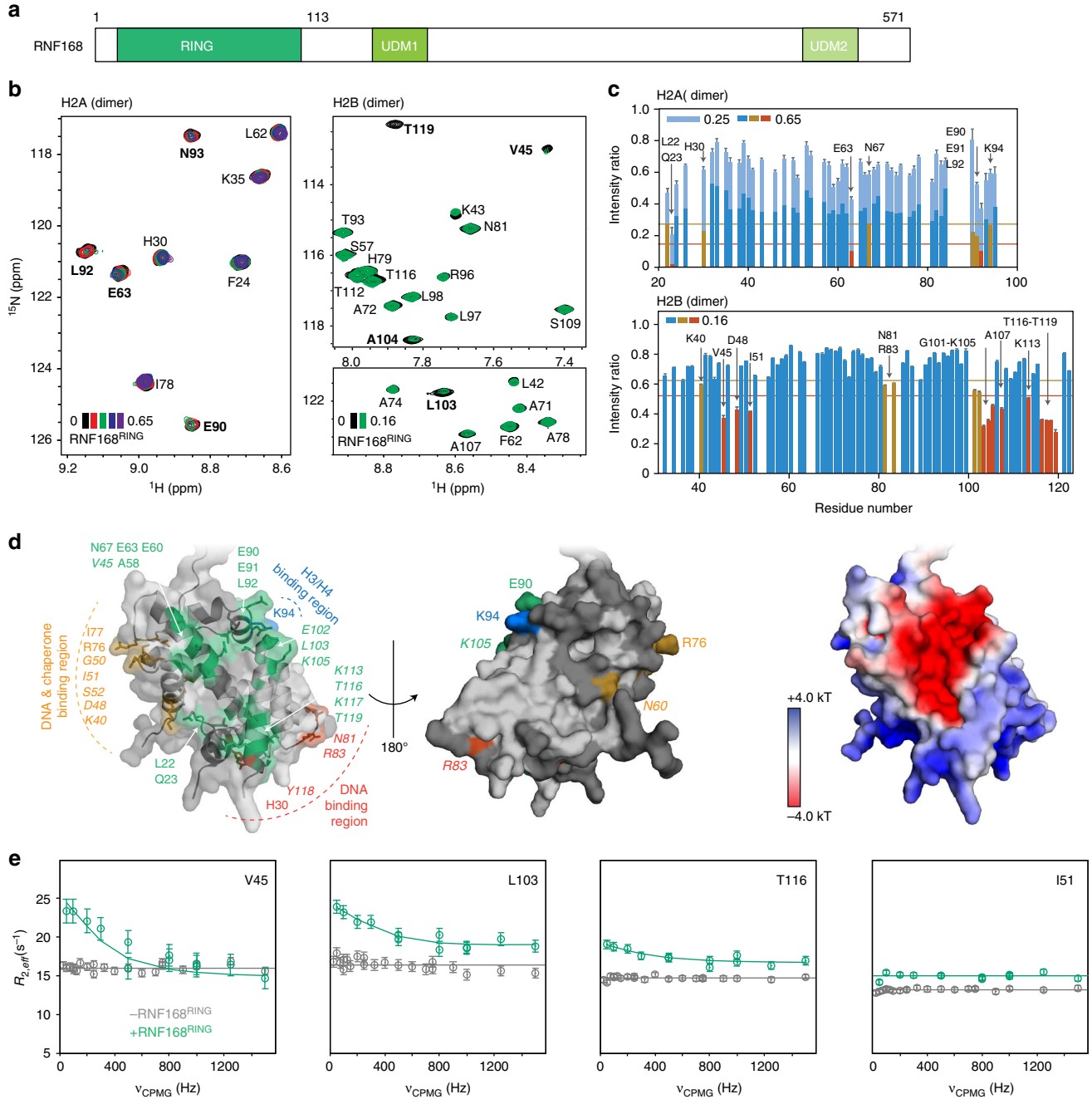

**Fig. 1** RNF168^RING binds to the acidic patch of H2A/H2B dimers. **a** Domain architecture of RNF168: the RING domain is required for ubiquitination, the UDM1 domain mediates recruitment to the DNA damage site through interaction with ubiquitinated H1 and the UDM2 domain binds ubiquitinated H2A-K15. **b** Sections of the 2D $^1$H-$^{15}$N correlation spectra of H2A (left) and H2B (right) within the H2A-H2B dimer with increasing amounts of RNF168^RING added. Color coding indicated, the number refers to molar equivalents RNF168 added compared to H2A/H2B dimer. Residues strongly affected by RNF168 binding are labeled in bold. **c** Normalized peak intensity ratios for H2A (top) and H2B (bottom) upon addition of sub-stoichiometric amounts of RNF168^RING (indicated in Figure in molar equivalents compared to H2A/H2B dimer). Residues with intensities that are one (two) standard deviation (SD) lower than the one-sided 10% trimmed mean are color coded in yellow (red) and labeled. **d** Residues with significant intensity/chemical shift changes (shown as sticks) cluster predominantly in and around the H2A-H2B acidic patch (see electrostatic view on the right), with additional effects observed for the regions otherwise occluded in the context of the nucleosome (indicated in orange, blue and red). H2B residue labels in italic. Residues in light/dark grey have no significant changes/no data available. **e** Addition of RNF168^RING results in clear exchange-induced line broadening for H2B residues in and around the acidic patch (V45, L103, T116) but not for residues in non-specific binding regions (I51) as evidenced by $^{15}$N CPMG relaxation dispersion. Color coding indicated in the figure. Best-fits (solid lines) for the acidic patch cluster are consistent with low micromolar dissociation constants and >1 ppm chemical shift changes. Error bars are s.d. based on noise levels in panel (**c**) and based on three replicate data points in panel (**e**). Source data are provided as a Source Data file

appreciably different response in NMR CPMG relaxation dispersion experiments. In these experiments the $^{15}N$ transverse relaxation rate, $R_{2,eff}$, is measured in a way that is very sensitive to dynamic chemical shift changes and larger chemical shift differences will cause larger dispersion of relaxation values. Addition of RNF168$^{RING}$ caused large dispersion in $R_{2,eff}$ values for V45, L103, and T116 that are located in the acidic patch area, but not for any of the resonances in other interfaces, such as I51 (Fig. 1e and Supplementary Fig. 1). Together with the absence of dispersion effects in the unbound state, this indicates that V45, L103, and T116 experience larger changes in chemical shift and/or have a higher population of the bound state than resonances such as I51. We thus conclude that RNF168$^{RING}$ binds the H2A/H2B acidic patch, and that in context of the dimer additional, non-specific binding modes are possible.

Next, these results were verified and extended by studying the interaction of the RNF168$^{RING}$ domain with the nucleosome, using methyl-based transverse relaxation optimized NMR (methyl-TROSY)[35]. Unlabeled RING domain was added to milligram quantities of nucleosomes, reconstituted from 601-DNA and fully deuterated histones from *Drosophila melanogaster*

(*Dm.*) in which the methyl groups of Ile, Leu and Val in H2A and H2B were $^{1}H$, $^{13}C$-labeled (ILV-labeling). The high solubility of the RNF168$^{RING}$-nucleosome complex allowed to perform the titration up to 2:1 molar ratio of RING domain to nucleosome. Significant and specific chemical shift perturbations (CSPs) can be observed for the methyl group resonances of L64 and L92 in H2A and V45 and L103 in H2B (Fig. 2a, b). These residues are part of or surround the acidic patch and thus reinforce the result from the dimer titration that the acidic patch is the specific binding site of the RNF168 RING domain (Fig. 2c). The shifting resonances are in the fast-exchange regime indicating a dynamic interaction with a ~1 ms upper limit for the life-time of the complex.

Previously, it was shown that the region encompassing a basic helix (residues 57–72) is required for binding and effective ubiquitination of H2A/H2B dimers and nucleosomes[9,23]. In particular, an R57D mutation was shown to interfere with nucleosome binding and ubiquitination of RNF168$^{RING}$. Interestingly, the basic-helix and its C-terminal loop contain additional basic residues R56, R63, R67, and R68 that could potentially mediate interaction with the acidic patch (see Fig. 2d)

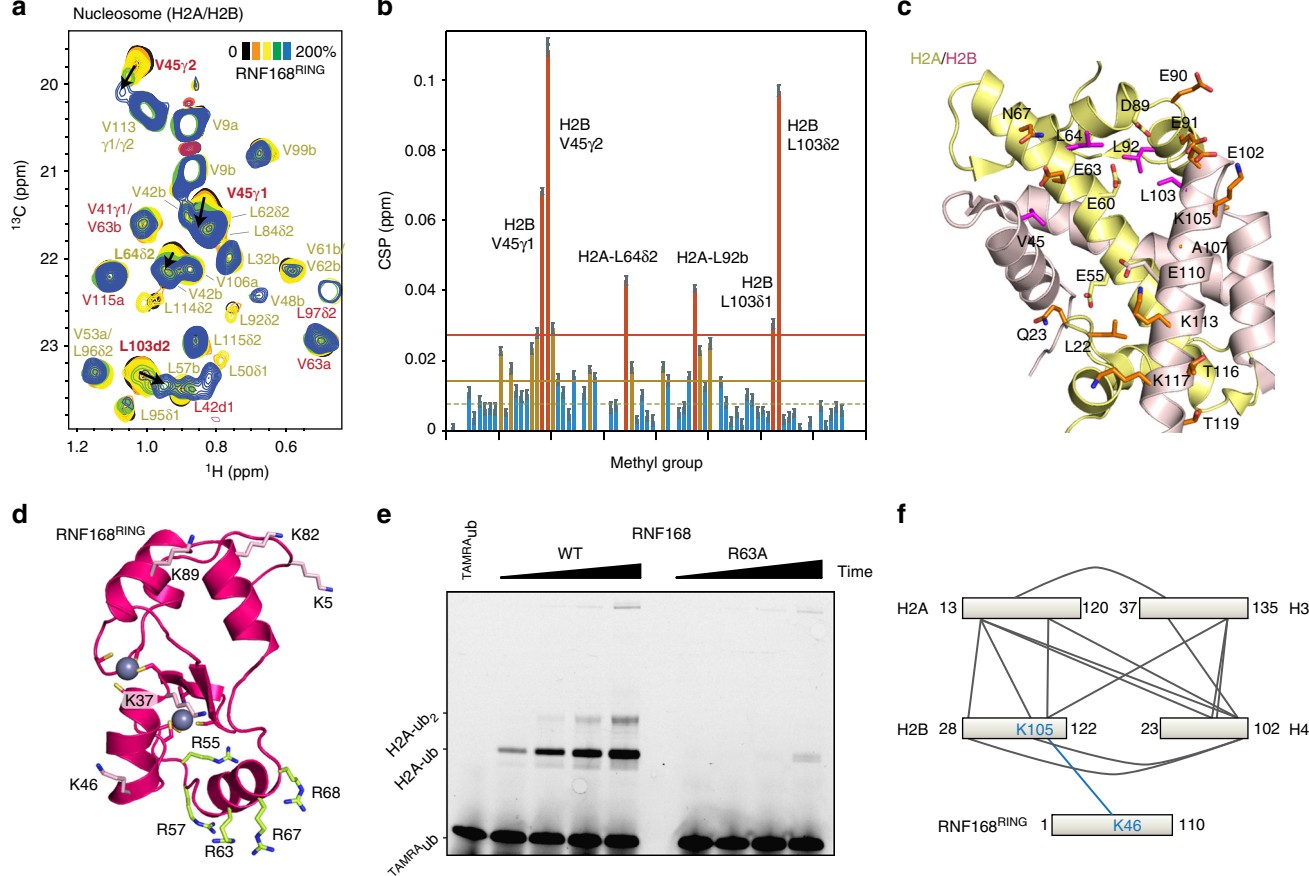

**Fig. 2** RNF168$^{RING}$ binds the nucleosome acidic patch using its Arg-rich helix. **a** Section of the 2D $^{1}H$-$^{13}C$ methyl-TROSY spectrum of the nucleosome with ILV-labeled H2A/H2B with increasing amounts of RNF168$^{RING}$ added. Color coding indicated. Labels a/b refer to either of the δ1/δ2 Leu or γ1/γ2 Val methyl groups. **b** Weighted chemical shift perturbation (CSPs) between the spectra of 1:2 and 1:0 nucleosome:RNF168$^{RING}$. Residues with CSPs that are one (two) standard deviations larger than the 10% trimmed mean are highlighted in yellow (red) and labeled. Error bars are s.d. based on 1 ppb standard error in peak position in $^{1}H$ and $^{13}C$ dimension. **c** The RNF168 binding surface as defined by NMR (shown as sticks, color coded in orange/magenta for dimer/nucleosome data). Side chains of other acidic patch residues are shown as sticks, color coded in yellow for H2A and light red for H2B. **d** Crystal structure of the RNF168$^{RING}$ domain with Arg residues of the basic-helix highlighted as yellow sticks. Lysine residues are shown as sticks and labeled, Zn atoms shown as spheres, coordinating residues as sticks. **e** Nucleosome ubiquitination by either wild-type (WT) or R63A mutant RNF168$^{RING}$. Fluorescent TAMRA-labeled ubiquitin was used to detect ubiquitinated species. Samples were resolved using SDS-PAGE. For both wild-type and mutant time points were taken at 5, 15, 30 and 60 min of incubation of the reaction mixture. Molecular size of mono- and di-ubiquitinated H2A is indicated. **f** Schematic representation of inter-histone crosslinks (gray; for rigid histone core only) and intermolecular RNF168-nucleosome crosslinks (blue) obtained from XL-MS. Source data are provided as a Source Data file

by acting as an arginine-anchor to bind the acidic patch[36]. Because of its central position in the putative nucleosome binding region, we tested nucleosome binding and ubiquitination activity of an RNF168 R63A mutant RING. While the mutant protein is folded, electrophoretic mobility shift assays (EMSA) indicate it binds nucleosomes with reduced affinity and in an altered binding mode (Supplementary Fig. 2). Compared to wild-type RNF168^RING, the R63A mutant conjugated only trace levels of fluorescently labeled ubiquitin to nucleosomes (Fig. 2e). Notably, we can rule out that the mutation caused ubiquitination of other histones since the readout in this experiment is the fluorescent ubiquitin itself.

To gain further insight in the nucleosome binding mode of RNF168^RING, we subjected a mixture of RNF168^RING and nucleosomes to crosslinking mass-spectrometry (XL-MS). The RING domain of RNF168 contains five lysine residues, of which two (K37 and K46) are relatively close to the Arg-rich helix (Fig. 2d). Using the amine-reactive bis(sulfosuccinimidyl)suberate as a cross-linking agent, a total of 62 unique crosslinks were obtained, mostly from lysines in the flexible histone tails (Fig. 2f and Supplementary Fig. 3). In total 21 crosslinks were found for the rigid histone core, the far majority in excellent agreement with the nucleosome crystal structure. Importantly, one intermolecular crosslink between RNF168 and the nucleosome was observed in all three replicate measurements. This crosslink links RNF168 K46, flanking the Arg-rich helix, to H2B K105, which lines the acidic patch (see also Fig. 2c). Together, mutagenesis, NMR, and XL-MS data indicate that RNF168^RING domain binds to the acidic patch using its arginine-rich helix.

**Integrative structure of the RNF168-nucleosome complex.** To achieve a deeper understanding of the underlying molecular mechanism of RNF168-mediated H2A ubiquitination, we sought to construct a structural model of the E3-nucleosome and E2-E3-nucleosome complex. On the basis of the experimental NMR, XL-MS, and mutagenesis data, we first docked RNF168^RING to the nucleosome using the data-driven docking program HADDOCK[37,38]. The crosslink was implemented as an unambiguous distance restraint between the Cα atoms of the cross-linked residues with 28 Å upper limit[39]. Since the docking calculation imposes this restraint as a Euclidian distance rather than a surface accessible distance (SASD), solutions with SASD > 35 Å were rejected. The final ensemble consists of one large and one smaller cluster of solutions (see Supplementary Fig. 4). The dominant cluster, shown in Fig. 3a, contains the overall best scoring solution and has more favorable physio-chemical scores. In this structure, the RNF168^RING basic helix is wedged between the H2A α2 and H2B αC helix and forms an extensive network of hydrogen bonding interactions with the acidic patch (Fig. 3b). RNF168 residue R63 is hydrogen bonded to H2A E60, D89, and E91, thus fulfilling the role of the canonical arginine-anchor[40]. R57 hydrogen-bonds to E60 and E63 and R67 provides additional interactions to D89 and E91. In addition, there are favorable interactions involving R56 and S60 in most structures, either by electrostatic interactions or hydrogen-bonding.

The Cα-Cα SASD between the crosslinked residues is 9.1 Å for the best scoring solution and 8.7 ± 1.1 Å for the best 10 structures, well within the typical limit of 30 Å[39]. We further validated the model by predicting all possible crosslinks (SASD < 30 Å)

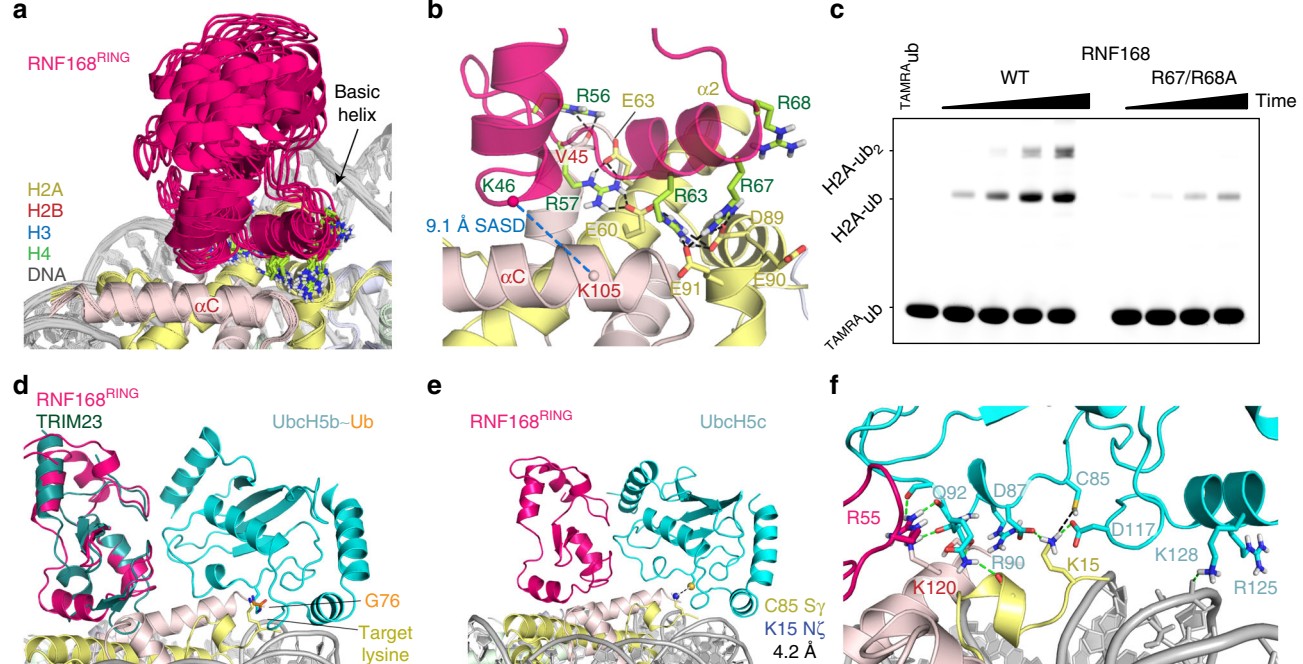

**Fig. 3** Structure of the RNF168^RING–nucleosome complex and implications for E2 positioning. **a** Superposition of the 10 best scoring solutions of cluster 1 (containing 187 out of 200 solutions) calculated using HADDOCK. Color coding indicated. Docking statistics are reported in Supplementary Fig. 4. **b** Zoom on the binding interface for the best scoring solutions of cluster 1, with key hydrogen-bond interactions involving the Arg-rich helix indicated. The Cα atoms and SASD of the crosslinked residues is indicated. **c** Nucleosome ubiquitination by either wild-type (WT) or R67/R68A mutant RNF168^RING. Fluorescent TAMRA-labeled ubiquitin was used to detect ubiquitinated species. Samples were resolved using SDS-PAGE. For both wild-type and mutant time points were taken at 2, 6, 18, and 40 min of incubation of the reaction mixture. Molecular size of mono- and di-ubiquitinated H2A is indicated. **d** Superposition of the TRIM23 E3 complex with Ub-conjugated E2 (PDB-ID 5VZW) and the RNF168^RING-nucleosome structure, showing close approximation of the Ub C-terminal residue G76 in the active site to the target lysine. **e** Structural model of the ternary UbcH5c-RNF168^RING-nucleosome complex calculated using HADDOCK. The best scoring solution is shown, distance between the E2 active site and target lysine is indicated. **f** Zoom on the UbcH5c-nucleosome interface showing hydrogen bonds with green dashes. Panels (**e** and **f**) show models with human histones, residues are numbered accordingly. Source data are provided as a Source Data file

between RNF168[RING] and the nucleosome based on the 10 best structures (see Supplementary Fig. 4). This yields five possible crosslinks, of which four pairs involve RNF168 K46. Experimentally, we only observed a single crosslink from RNF168 K46, to H2B K105. Since the SASD for this pair is much shorter than the other three, it suggests the observed crosslink is more readily formed, thus precluding the observation of the other predicted links. A fifth predicted crosslink is expected between RNF168 K37 and H2B K105 but this is not found. Again, H2B K105 may more readily crosslink to RNF K46 which is much closer in the structure (on average 8.7 vs. 22 Å SASD). We thus conclude that the model is in good agreement with experimental crosslinking data.

Since our structure predicts a direct role of RNF168 R67 in the interface, we performed an additional mutation study with an R67/68A double mutant to verify these results. Here, R68 was included in the mutation because of its proximity to R67. While the mutant protein binds with comparable affinity to nucleosomes as wild-type, the altered band-shift pattern indicates a change in binding mode (Supplementary Fig. 2). The mutant exhibits strongly decreased ubiquitination activity (Fig. 3c), suggesting the mutant has reduced ability to bind in a productive binding mode. As for the R63A mutant, we can exclude that ubiquitination activity is shifted to other histones than H2A. The residual activity observable is in contrast to the near inactive R57A[23] and R63A mutant (Fig. 2e). This correlates well with the more extensive hydrogen bonding by R57 and R63 compared to R67.

Crucially, ubiquitination target specificity is determined by interplay between E2, Ub-conjugated E2 (E2~Ub) and the substrate. We thus selected the E3-E2~Ub complex structure that most closely resembled RNF168[RING] E3 structure and superimposed this on our RNF168[RING]–nucleosome complex structure (Fig. 3d). Strikingly, this results in the placement of the E2 active site with the conjugated C-terminus of Ub directly above the target lysines. Since the superposition introduces a few steric clashes between the E2 and the nucleosome surface, we next used HADDOCK to model the ternary complex between E3 RNF168[RING] domain, E2 enzyme and the nucleosome (containing human histones). While the cognate E2 of RNF168 is yet unknown, we selected UbcH5c as it can function as such in vitro[9]. Restraints for the modelling were based on structurally conserved E3-E2 interactions, including the so-called linchpin hydrogen bond, which is crucial for E2 activation[6] (see Supplementary Fig. 5). In RNF168 this involves hydrogen-bond between R55 and the backbone of E2-Q92. Importantly, proximity of the E2 to the target lysines is not enforced during docking. Since the HADDOCK protocol allows for limited flexibility, clashes in the intermolecular interfaces can be removed and their physio-chemical quality optimized.

The resulting best scoring model places the E2 catalytic center in close proximity to the H2A K15 (4.2 Å Sγ-Nζ distance) without introducing steric clashes (Fig. 3e). Within the 20 best solutions, the closest distance is 3.3 Å. The second lysine ubiquitinated by RNF168, H2A K13 is not present in the model. Both from the absence of electron density in many crystal structures and from recent NMR studies on the nucleosome[34], it is clear that this region of the H2A N-terminal tail is dynamic. Superposition with nucleosome structures that include this side chain shows that K13 is still reasonable close such that with some backbone and sidechain reorientation it can get close enough to the active site to be ubiquitinated (Supplementary Fig. 6).

A lysine in the H2B αC-helix, *Hs*. H2B K120 that is ubiquitinated by E3 ligase Bre1 (RNF20/40)[41,42], is relatively close to the H2A K15 site. The distance from this lysine to the UbcH5c catalytic center in our structure is however 15.7 Å, thus

explaining the preferential ubiquitination of the H2A N-terminus by RNF168. This specificity is well maintained over the best 20 solutions, with no solutions that place H2B K120 within 9 Å. The C-terminal H2B residue, *Hs*. H2B K125, is similarly placed at 14.8 Å in the best solution with no solutions within 6 Å.

Analysis of the E2–nucleosome interface in the 20 best scoring solutions shows that there are few intermolecular interactions between the E2 and the nucleosome, primarily via a hydrogen-bond between H2A K15 and D117, the so-called gateway residue[43] (Fig. 3f). In addition, most solutions show an interaction to H2B K120 and in some solutions the E2 interacts with the nucleosomal DNA through K128 or R125. An NMR titration of UbcH5c to H2A/H2B dimers in presence of sub-stoichiometric amounts of RNF168 did not reveal a clear binding interface for the E2, likely due to a lack of stable complex formation (Supplementary Fig. 7). E2 binding may be stabilized in the nucleosomal context, which is also suggested by the higher activity of RNF168/UbcH5c on nucleosomes compared to dimers[23].

Overall, our data-driven integrative structure of the E3-nucleosome complex and structural model of the E2-E3-nucleosome complex indicate that RNF168 relies on multiple critical arginine-acidic patch interactions to bind the nucleosome in a productive conformation, thereby directing the E2 towards the target lysine.

**Uncoupling of PRC1 and RNF168 ubiquitination.** We find that the RING domain of RNF168 binds to the acidic patch on the nucleosome surface, the very same binding epitope as was found for another E3 RING domain, the heterodimeric RING1B/BMI1[12]. These two E3 have their respective target lysines, K13/15 for RNF168 and K118/119 for RING1B/BMI1, at opposite sides of the nucleosome with respect to the acidic patch. The opposite position of the target lysines correlates directly with the approximate 180° turn of the E2 between both complexes, caused by a rotation of the E3 RING domain on the nucleosome surface (Fig. 4a, b). The opposite N-to-C terminal direction of the nucleosome binding motifs of RNF168[RING] and RING1B is emphasized in Fig. 4b. The different binding modes of RNF168 and RING1B/BMI1 on the nucleosome suggest the possibility to selectively manipulate their interactions and subsequent ubiquitination activity. Since the binding site of the two E3 proteins overlap extensively in the anchor region, we hypothesized that mutation of histone residues in the periphery of the RNF168 interface that are distant from the RING1B interface could be effective in selective silencing of RNF168-mediated ubiquitination. In addition, the mutagenesis experiments suggest that interference with the electrostatic environment of RNF168 could be effective, as productive RNF168 binding proved highly sensitive to mutations of arginines in and around the basic helix. We thus chose H2B E110 (E113 in humans) as a feasible mutation site as it is in the periphery of the basic-helix interaction site, in the proximity of R57, and neither interacts with RING1B nor is required for the structural integrity of the H2A/H2B-dimer (Fig. 4b). Indeed, when wild-type nucleosome or nucleosomes reconstituted from a H2B E110A mutant are interrogated for both types of ubiquitination, H2A ubiquitination mediated by RING1B is unaffected while RNF168-dependent ubiquitination is very strongly decreased (Fig. 4c). These results demonstrate that it is possible to selectively interfere in ubiquitination pathways by mutations far from the target lysine.

**Discussion**
RNF168 is an important driver of the DNA damage response, an activity that depends on its RING domain-mediated

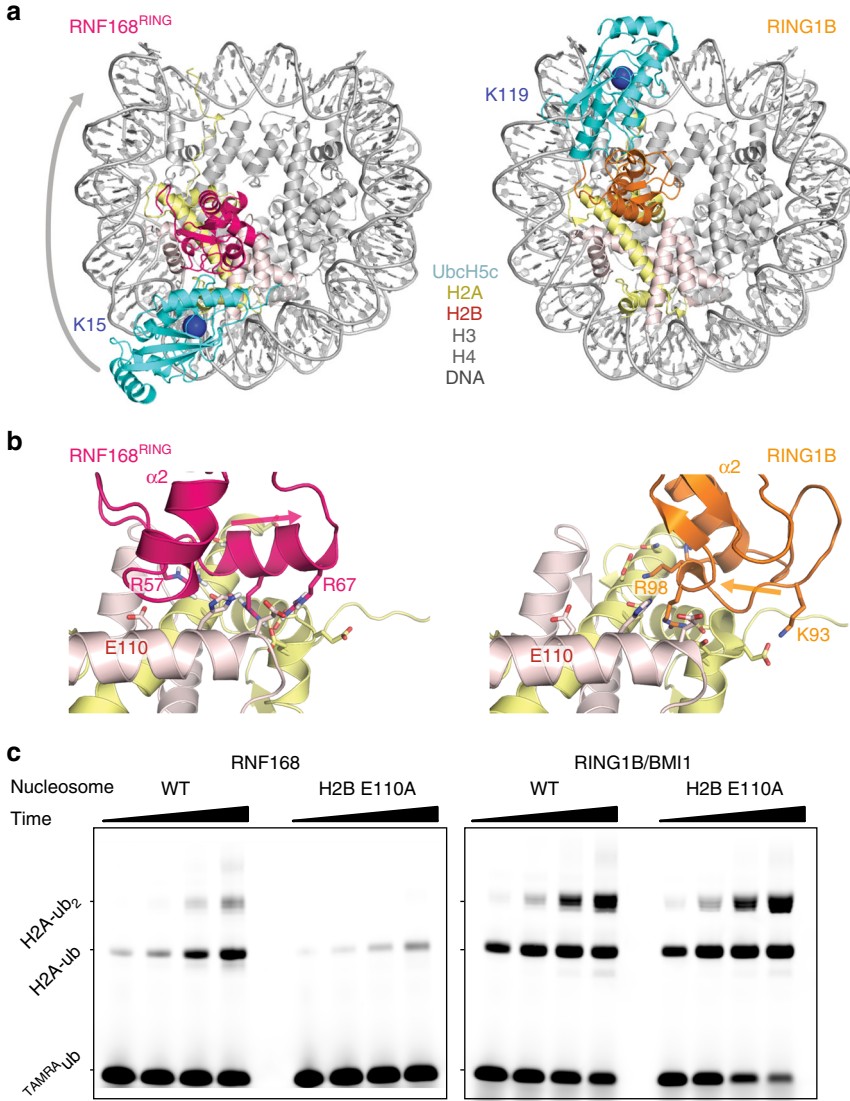

**Fig. 4** Uncoupling of PRC1 and RNF168 ubiquitination. **a** Top view on the nucleosome highlighting the 180° rotated position of UbcH5c in the RNF168 (left) compared to RING1B nucleosome complex (right). Color coding indicated in the Figure, position of the target lysines is indicated with the blue sphere. **b** Side-by-side comparison of the RING domain positions on the acidic patch, RNF168 (right) and RING1B (left). Secondary structure elements are labeled to highlight the approximate 180° turn of the RING domain, also indicated by the arrow. The H2B E110 (E113 in humans) side chain is close to the E3 RING surface for RNF168 but far from the surface of RING1B. Selected side chains are shown as sticks. **c** Nucleosome ubiquitination by either RNF168$^{RING}$ or RING1B/BMI1 on wild-type (WT) or H2B E110A nucleosomes. Fluorescent TAMRA-labeled ubiquitin was used to detect ubiquitinated species. Samples were resolved using SDS-PAGE. For all samples time points were taken at 2, 6, 18, and 40 min of incubation of the reaction mixture. Molecular size of mono- and di-ubiquitinated H2A is indicated. Source data are provided as a Source Data file

ubiquitination of H2A K13/K15. Here, we used a combination of methyl-based and amide backbone-based NMR to show that RNF168$^{RING}$ binds the H2A/H2B acidic patch on the nucleosomal surface. Based on NMR, XL-MS, and mutagenesis data, we derived a data-driven structural model of the E3-substrate complex formed by RNF168$^{RING}$ and the nucleosome detailing the interaction between the Arg-rich helix of RNF168 and the acidic patch. Recent studies have shown that one of the UDM motifs of RNF168 also bind the acidic patch to aid recognition of the H2A-K15Ub mark[26,27], making RNF168 the first example of a histone modifier in which both writing and reading modules bind the same epitope on the nucleosome, yet distant from the modification itself.

Our structure shows how the binding mode of the E3 directs and positions the E2 towards the target lysine, thereby explaining the observed specificity of H2A ubiquitination. The observed

crosslink between RNF168 and the histone surface has been crucial in defining the E3 binding mode. NMR and mutagenesis data identify binding surfaces only and thus do not directly encode in which orientation these surfaces interact. The observed crosslink established unambiguously the orientation of RNF168 on the nucleosome surface, thus resulting in a single binding mode. Importantly, the structure is validated by additional mutagenesis, and provided a structural guide to design a nucleosome mutant that selectively interferes with K13/15 but not K118/119 ubiquitination.

We find that the RNF168$^{RING}$ basic helix is anchored by three arginine residues, R57, R63, and R67, to the acidic patch, while previous work had failed to identify a direct interaction to the acidic patch[23]. The histone mutant used in this study included a mutation for acidic patch residue E90, which is pointing away from the interface in our model, but not for D89 and E91, which

are bound to both R63 and R67, and thus likely failed to abolish the interaction.

The same work by Mattiroli et al. established that the nucleosome promotes the rate of ubiquitination discharge[23], thus uncovering a catalytic role of the nucleosome substrate. To shed light on the structural underpinnings of this striking observation, we compared our structural model of the E2-E3-nucleosome complex with recent E3-E2~Ub structures. These structures have highlighted the importance of Ub–E2 and Ub–E3 interactions in stabilizing the closed, active state of the complex. In this state the Ub C-terminal tail is bound by the E2 and thought to be strained, thus activating the thioester bond between C-terminus of Ub and the catalytic cysteine of the E2[4–6]. Alternatively, the complex may be in an open state, in which the Ub is dynamically sampling a large conformational space. Superposition of such open E3-E2~Ub structures onto our model of the E2-E3-nucleosome complex shows that these states are incompatible with nucleosome binding due to severe clashes of Ub with the nucleosome (Fig. 5a). Superposition of E3-E2~Ub complexes in the closed state with our model shows that our structure is compatible with such closed states and the formation of E2-Ub contacts (Fig. 5b). Notably, the orientation of RNF168[RING] on the nucleosome surface is fully compatible with the formation of E3-Ub contacts that are observed in the structure of related RING domains bound to E2~Ub (Supplementary Fig. 6). We thus suggest that the nucleosome itself, by steric occlusion, constrains the conformational space of Ub in the E3-E2~Ub complex and thus promotes the formation of closed complexes.

Compared to the PRC1 machinery responsible for H2A K119 ubiquitination, we find that the E3 RNF168[RING] domain is rotated ~180° on the nucleosome surface, explaining the difference in target specificity (see Fig. 4). This raises the question as to the molecular origin of this this difference in binding mode. Simple considerations underline the evident need for a specific binding mode for each E3. First, the helical structure of this element in RNF168[RING] would cause steric clashes with the nucleosome surface when adopting the exact same binding mode as RING1B (Fig. 6a). Second, the basic surfaces of two RING domains are oriented differently with respect to the E2 binding site and have different shapes and charge distributions (Fig. 6b). In particular, the area around the linchpin Arg, R91 in RING1B and R55 in RNF168, is much less electropositive in RING1B than

in RNF168 due to the presence of two glutamic acid residues at the surface. At the opposite side of the molecule, RNF168 is less electropositive, mainly due to the presence of E45. The overall result is that the basic surface of RNF168 is rod-shaped while that of RNF168 is rather square shaped.

Examination of the two complexes shows how these differences correlate with the observed binding mode (Fig. 6c). The positive extension of RNF168 interacts with region around H2A E55, thus filling the negatively charged groove formed by the H2A α2 and H2B αC helix completely. In RING1B the area around H2A E55 is not contacted, instead the unique extension of RING1B, R81, one of the drivers of nucleosome binding[12], interacts outside the α2-αC cleft with H2A D72. In RNF168, the presence of E45 around this position favors interaction with H2B K105 on the opposite side of the cleft. It is thus tempting to speculate that this combination of changes, i.e. a linear vs. square shaped extension of a core basic surface around the arginine anchor, is responsible for the change in binding mode. Experiments in which the nucleosome-binding elements of the two E3 RING domain were swapped resulted in inactive protein. A further complicating factor is that RING1B binds as a heterodimer with BMI1 to the nucleosome surface, with BMI1 predominantly contacting H3/H4[12]. Nevertheless, the differences in RING-nucleosome interaction between RING1B and RNF168 highlight the importance of residues surrounding the arginine anchor in determining the exact binding mode.

The RNF168[RING]–nucleosome structure was used to design an H2B E110A nucleosome mutant to selectively suppress RNF168 activity without interfering with RING1B mediated ubiquitination. The successful design further validates the structural model. Since RNF168 still shows low residual activity on nucleosomes carrying the H2B E110A mutant, a charge reversal at this position may be needed to completely and selectively silence RNF168. This H2B E110 mutant (E113 in humans) may be an excellent tool to investigate both RNF168 and PRC1 pathways without interfering with the target lysine residue.

Overall, our results contribute to a better understanding of the molecular mechanism of nucleosome ubiquitination, underscoring the crucial and diverse roles that RING type E3 ligase and their substrates play in ubiquitination. Our data also highlight how chromatin factors can exploit the nucleosome acidic patch in very different ways to control chromatin biology and cell fate.

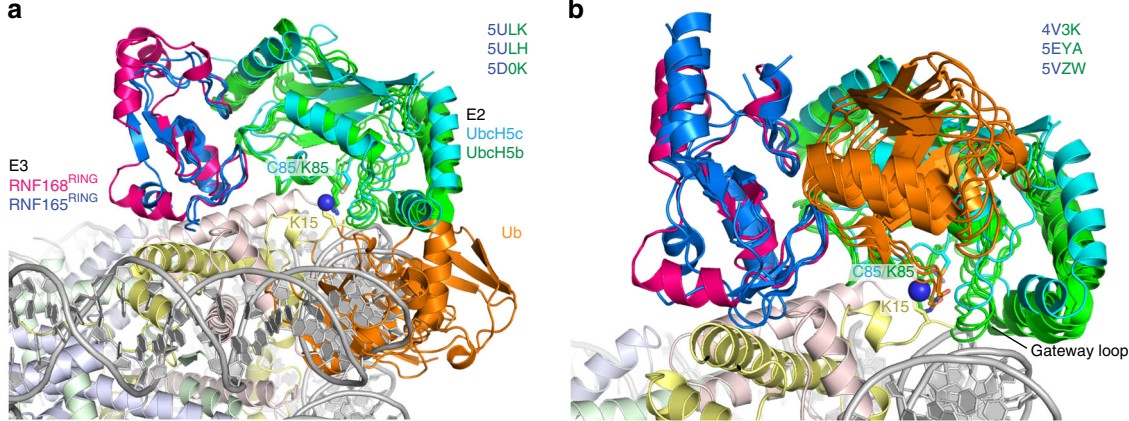

**Fig. 5** The nucleosome promotes active, closed E3-E2~Ub conformations. **a** Superposition of E3-E2~Ub structures in the open state with E3 RNF165 and the E2-E3-nucleosome model for RNF168 indicates that nucleosome surface clashes with such open states. **b** Superposition of RING E3-E2~Ub structures and the E2-E3-nucleosome model for RNF168 shows that the model is compatible with the closed state of the E3-E2~Ub complex. There are no direct interactions between the nucleosome surface and Ub. Target lysine, here labeled as K15, catalytic center, and gateway loop are indicated. Color coding indicated in the Figure

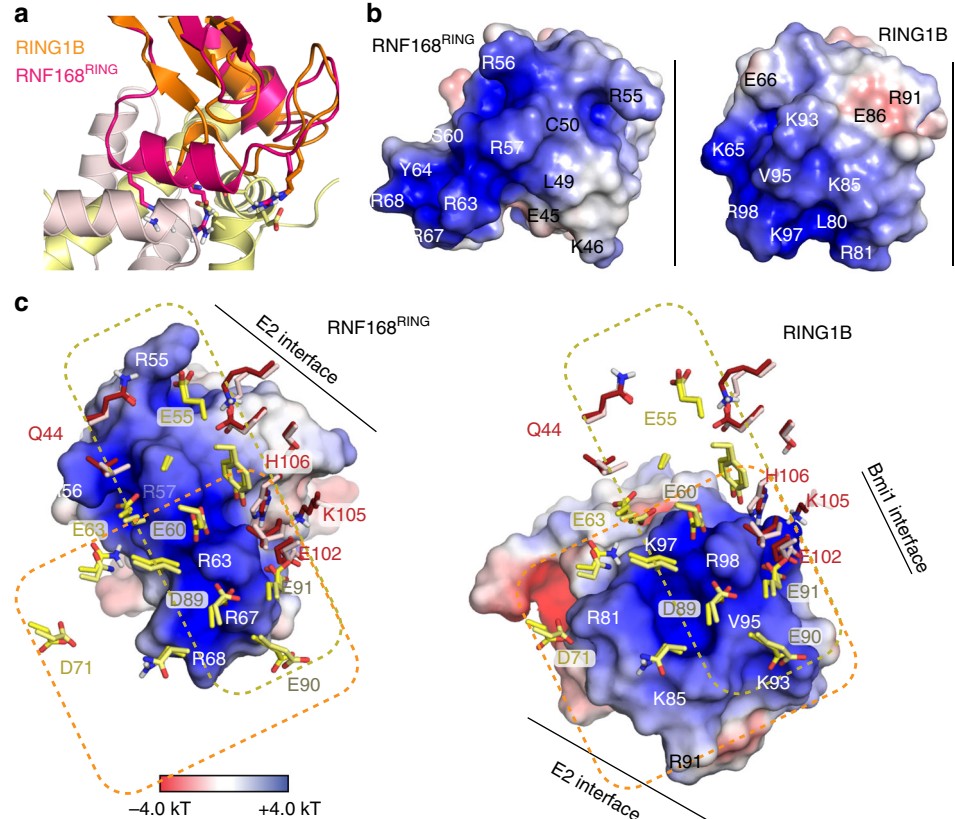

**Fig. 6** Comparison of acidic patch binding mode of RNF168 and RING1B. **a** Superposition of RNF168[RING] on the RING1B structure results in severe clashes of the Arg-rich helix with the H2B αC helix. **b** Molecular surface of the bound states of RNF168 and RING1B RING domains with color coded electrostatic potential, calculated for the solvent accessible surface (color scale shown in panel **c**). Both RING domains are shown looking down on their nucleosome binding surface in an aligned orientation. Location of the E2 interface is indicated with a solid line. **c** Comparison of the acidic patch binding modes of RNF168[RING] and RING1B[RING], both shown as molecular surface with color coded electrostatic potential, calculated for the solvent accessible surface, and aligned on the nucleosome. The yellow and orange dashed boxes delineate the main interaction sites of RNF168 and RING1B. Selected residues are labeled

## Methods

**Protein expression and purification.** For NMR studies, *Drosophila melanogaster* histones were expressed in *E. coli* BL21 Rosetta2 (DE3) cells (Novagen) and purified under denaturing conditions from inclusion bodies by extraction in 6 M guanidium chloride, followed by size-exclusion chromatography in buffer A (7 M urea, 50 mM NaPi, 1 mM EDTA, 150 mM NaCl, pH7.5) using a Superdex 200 column (GE) and ion exchange with a salt gradient from buffer A to buffer A with 1 M NaCl.[44] For the ubiquitination, gel-shift and crosslinking studies *Xenopus leavis* histones were used. Histones used for NMR studies were produced in M9 minimal medium containing desired isotopes, histones used for ubiquitination assays were expressed in LB medium. Methyl-labeling of Ile-δ1-[$^{13}CH_3$] and Val/Leu-[$^{13}CH_3$, $^{12}CD_3$] (ILV-labeling) was achieved by adding 60-80 mg of labeled precursor (CIL) to the expression medium 1 h before induction[35].

The RNF168-RING domain construct (residues 1–113) was cloned into petNKI-His-SUMO2-kan vector and purified using a His-SUMO tag. Mutations were applied using the Quickchange site-directeced mutagenesis kit (Agilent). Plasmids for wild type and mutant proteins are available from the authors. Proteins were expressed in *E. coli* BL21 Rosetta2 (DE3) cells (Novagen). Freshly transformed cells were grown to an OD of 0.6 and induced with 200 μM IPTG. The protein was expressed overnight at 16 °C. The cells were harvested in lysis buffer (50 mM HEPES pH 7.5, 150 mM NaCl, 1 μM ZnCl₂, 1 mM TCEP, 2 mM imidazole) including complete EDTA-free protease inhibitor. Lysate was cleared by centrifuging at 21000 g and the supernatant was loaded on chelating sepharose beads (GE healthcare) charged with Ni²⁺. The beads were washed with lysis buffer containing 20 mM imidazole and the protein was eluted using 350 mM imidazole. The His-SUMO tag was cleaved overnight using His-tagged SENP2 protease while dialyzing against 50 mM HEPES pH 7.5, 150 mM NaCl, 1 μM ZnCl₂, 1 mM TCEP. Uncleaved protein and SENP2 protease were removed using chelating sepharose beads charged with Ni²⁺. The sample was then diluted to 50 mM NaCl and loaded onto a Heparin column (GE healthcare). The protein was eluted using a salt gradient ranging from 50 mM to 1 M NaCl over 12 column volumes. Fractions containing RNF168 were combined and gel filtered on a Superdex 75 16/60 column (GE healthcare) in 25 mM HEPES pH 7.5, 150 mM NaCl, 1 μM ZnCl₂, 1 mM TCEP. Fractions containing RNF168 were combined, snap frozen in liquid nitrogen and stored at −80 °C.

**Histone refolding and nucleosome reconstitution.** Histone H2A/H2B dimers or histone octamers were refolded from equimolar mixes of denatured purified histones by dialysis to 2 M NaCl and subsequent purification using size-exclusion chromatography over an Superdex 200 column (GE)[44]. The 601-DNA Widom-template was produced from a pUC19 plasmid containing 12 copies of the 601 fragment amplified in *E. coli* DH5α (Novagen) and extracted by alkaline lysis followed by isopropanol and ethanol precipitations[45]. The pellet was dissolved in TE buffer (20 mM Tris-HCl pH 7.5, 5 mM EDTA, 100 mM NaCl) and purified by anion exchange chromatography. The purified plasmid was restricted with ScaI (ThermoFisher) and purified by anion exchange chromatography. Nucleosomes were reconstituted by preparing an equimolar mix of purified 167 bp 601 DNA and refolded histone octamers at 2 M NaCl followed by salt-gradient dialysis to 0.25 M NaCl[45].

**Fluorescent labeling of ubiquitin.** Ubiquitin carrying a cysteine as residue 2 was cloned into the petNKI-His-SUMO2-kan vector and expressed in *E. coli* BL21 Rosetta2 (DE3) cells (Novagen). Cells were grown in LB until an OD of 0.8 was reached and then induced with 200 μM IPTG. The protein was expressed for 4 h at 37 °C and cells were harvested in lysis buffer (50 mM Tris pH 7.5, 150 mM NaCl, 1 mM TCEP and 5 mM imidazole). Cells were lysed by sonication and the lysate was cleared by centrifuging at 210,00 × g. The supernatant was loaded on chelating sepharose beads charged with Ni²⁺ and washed with lysis buffer containing 20 mM imidazole. The protein was eluted in lysis buffer containing 350 mM imidazol. The His-SUMO tag was cleaved over night with SENP2 protease while dialyzing against 50 mM Tris pH 7.5, 150 mM NaCl, 1 mM TCEP. Protease, uncleaved protein and His-SUMO tag was removed using chelating sepharose charged with Ni²⁺. Perchloric acid was added dropwise to the sample wile stirring on ice until a final concentration of 2% v/v. The sample was centrifuged at 21,000 × g and the supernatant was collected and dialyzed against 50 mM ammonium acetate pH 4.5 overnight. The sample was loaded on a SP HP column (GE healthcare) and eluted using a linear salt gradient ranging from 0 to 500 mM NaCl. The sample was then gel-filtered in 50 mM Tris pH 8, 100 mM NaCl, 5 mM DTT and stored at − 80 °C. Before labeling the required amount of ubiquitin was dialyzed against at least three changes (one step overnight) of 2 L labeling buffer (50 mM Tris, pH 7.5, 150 mM NaCl) to remove the DTT. The ubiquitin was then labeled by adding a five-fold

molar excess of TAMRA-maleimide (Setareh Biotech). Samples were incubated for 2 h at room temperature, followed by 14 h at 4 °C. The labeling reaction was quenched by adding 5 mM DTT and excess label was removed through size-exclusion chromatography using a Superdex S75 column (GE healthcare) in 50 mM Tris pH 7.5, 150 mM NaCl. The labeled protein was stored at −80 °C

**Ubiquitination assays**. All ubiquitination assays were done in a buffer containing 50 mM Tris pH 7.5, 150 mM NaCl, 5 mM MgCl$_2$, and 5 mM DTT. Nucleosomes were ubiquitinated using 1 µM hUBA1, 0.5 µM UBE2D3, 1 µM RNF168 (wild type or the respective mutant), 7 µM recombinant nucleosome core particles containing *Xenopus laevis* histones, 15 µM TAMRA-labeled ubiquitin at 30 °C. The reaction was started by adding 3 mM ATP and stopped by mixing with SDS-loading dye at the desired time point. Ubiquitinated histones were resolved using SDS-PAGE. Bands were visualized using fluorescence of TAMRA-labeled ubiquitin. Gel images were cropped and adjusted for contrast and brightness using the levels tool in Photoshop. Uncropped gel images are available in the Source Data file.

**Electrophoretic mobility shift assays**. Recombinant nucleosome core particles containing *Xenopus laevis* histones (250 nM) were mixed with varying concentrations of RNF168$^{1-113}$ (wild-type or the respective mutants) in a buffer containing 50 mM Tris-HCl, pH 7.5, 150 mM NaCl, 1 mM DTT. Samples were incubated at room temperature for 20 min and then separated using a 0.2 × TBE, 6% Acrylamide native gel run at 120 V at 4 °C for 75 min. The gel was pre-run prior to the experiment for 90 min at 4 °C at 120 V. The gel was then stained using SYBR-safe stain (ThermoFisher) and Coomassie Brilliant Blue. Gel images were cropped and adjusted for contrast and brightness using the levels tool in Photoshop. Uncropped gel images are available in the Source Data file.

**NMR spectroscopy**. All NMR experiments were carried out at 293 K on a Bruker Avance III HD spectrometer operating at 850 MHz $^1$H Larmor frequency equipped with a TCI cryo-probe, unless noted otherwise. Processing was done using the NMRPipe package[46] or Bruker's TopSpin. Spectra were analyzed using Sparky (Goddard and Kneller, UCSF).

Samples for assignment of H2A contained ~300 µM [U-$^2$H/$^{13}$C/$^{15}$N]-H2A-H2B or [U-$^{13}$C/$^{15}$N]-H2A-H2B in 95/5% H$_2$O/D$_2$O in NMR buffer (20 mM NaPi, pH 6.5, 50 mM NaCl, 5% D$_2$O, 0.02% NaN$_3$, complete EDTA-free protease inhibitor cocktail (Roche)). Backbone assignments of H2A in the H2A/H2B dimer were based on TROSY-based HNCACB, HN(CO)CACB, HNCA, HN(CO)CA, HNCB, HN(CO)CB, HNCO, and HN(CA)CO spectra, recorded at 308 K. Overall assignment completeness was 97.1% for all backbone atoms. Assignments are deposited in the BMRB databank under accession code 27547.

NMR titration experiments of H2A-H2B histone dimer with RNF168$^{RING}$ were performed using ~110 µM [U-$^1$H,$^{15}$N]-H2A-H2B for the H2A-observed titration and ~200 µM [U-$^2$H]-H2A-[U-$^2$H,$^{15}$N]-H2B for the H2B-observed titration and ~460 µM unlabeled RNF168$^{RING}$ stock. Both proteins were extensively dialyzed to 90/10% H$_2$O/D$_2$O with 25 mM Tris pH 8.0, 100 mM NaCl, 10 µM ZnCl$_2$ and 3 mM DTT. Nucleosome interaction studies were done at 298 K, using ~25 µM mononucleosomes reconstituted with ILV-H2A, ILV-H2B/$^2$D-H3 & $^2$D-H4 labelling scheme. The buffer conditions were 30 mM Tris pH* 7.3 in 100% D$_2$O with 100 mM NaCl, 1 mM TCEP and 1 µM ZnCl$_2$. Methyl group assignments were transferred from the original assignment condition[29] to the interaction buffer using a buffer and temperature titration.

The titration and CPMG relaxation dispersion study of H2B-labeled H2A/H2B dimers were carried on a Bruker Avance III HD spectrometer operating at 950 MHz $^1$H Larmor frequency equipped with a TCI cryo-probe. Relaxation dispersion experiments were recorded on the $^{15}$N TROSY coherence[47] using a constant-time CPMG relaxation delay of 40(20) ms and 22(13) $\nu_{CPMG}$ values, including three replicates for error estimation, in the range of 25(50)−1500 Hz for the unbound (bound) dimer. Peak volumes were determined using Fuda[48] and best-fits were obtained using Catia[49]. CSPs were calculated as average weighted perturbations in ppm, using a weighting factor based on standard deviation of chemical shifts in the BMRB, i.e., 1 and 0.15 for the shifts in the $^1$H and $^{15}$N dimension or 1 and 0.32 for shifts in the $^1$H and $^{13}$C dimension.

**Crosslinking mass-spectrometry**. Nucleosome core particles (NCP), reconstituted from *Xenopus laevis* histones and 601-DNA, and RNF168$^{1-113}$ were crosslinked at final concentrations of 15 µM RNF168 and 5 µM NCP in 10 µL volume using a buffer containing 25 mM HEPES pH 7.5, 150 mM NaCl. The samples were incubated for 10 min at room temperature. A 500-fold molar excess of BS3 (with respect to NCP concentration) was added and crosslinking proceeded for 20 min at room temperature. The reaction was stopped by addition of Tris pH 8 to a final concentration of 125 mM. The samples were then diluted to a volume of 100 µL using a buffer containing 50 mM Tris pH 8 and 150 mM NaCl. TCEP was added to a final concentration of 10 mM and the samples were incubated at 60 °C for 30 min. Chloroacetamide was added to a final concentration of 40 mM and the samples were incubated in the dark for 20 min. The samples were then digested using trypsin at 37 °C overnight. The digest was stopped by adding formic acid to a final concentration of 1%. The digested samples were purified using 100 µl ZipTip pipette tips (Merck) according to the manufacturer's instructions. Samples were

then concentrated to ~5 µL using a SpeedVac vacuum centrifuge and diluted with 20 µL Buffer A (0.1% formic acid).

The peptides were analyzed by online nano-high-pressure liquid chromatography (UPLC) electrospray ionization-tandem mass spectrometry (MS/MS) on an Q Exactive Plus Instrument connected to an Ultimate 3000 UPLC (Thermo-Fisher Scientific). Peptides reconstituted in 0.1% formic acid were loaded onto a trap column (Acclaim C18 PepMap 100 nano Trap, 2 cm × 100 µm I.D., 5-µm particle size and 300-Å pore size; Thermo-Fisher Scientific) at 15 µL/min for 3 min before switching the precolumn in line with the analytical column (Acclaim C18 PepMap RSLC nanocolumn, 75 µm ID × 50 cm, 3-µm particle size, 100-Å pore size; Thermo-Fisher Scientific). The separation of peptides was performed at 250 nL/min using a non-linear ACN gradient of buffer A (0.1% formic acid) and buffer B (0.1% formic acid, 80% ACN), starting at 2.5% buffer B to 42.5% over 95 min. Data were collected in positive mode using a Data Dependent Acquisition $m/z$ of 375–2000 as the scan range, and higher-energy collisional dissociation (HCD) for MS/MS of the 12 most intense ions with $z$ 2–5. Other instrument parameters were: MS1 scan at 70,000 resolution, MS maximum injection time 118 ms, AGC target 3E6, ion intensity threshold of 4.2e4 and dynamic exclusion set to 15 s. MS/MS resolution of 35000 at Orbitrap with the maximum injection time of 118 ms, AGC of 5e5 and HCD with collision energy = 27%.

For the data analysis, Thermo raw files were analyzed using the pLink 2.3.4 search engine[50], searching against the sequences of RNF168$^{1-113}$, Histone H2A, H3, H4, and H2B in FASTA format. The default settings for searches were used. N-terminal acetylation and methionine oxidation were used as variable modifications and carbamidomethyl on cysteines as a fixed modification. False discovery rates of 1% for peptide spectrum match level were applied by searching a reverse database. Reproducible crosslinks were identified from three replicate experiments and manually verified. Crosslinks within the histone core were analyzed for compatibility with nucleosome structure by calculating the solvent accessible surface distance using Jwalk[39].

**Structural modelling**. The structural model of the RNF168$^{RING}$-nucleosome complex was determined using the experimental NMR, XL-MS, and mutagenesis data in the data-driven docking software HADDOCK[37]. Briefly, the RNF168$^{RING}$ domain (PDB-id 4GB0[51]) was docked to the nucleosome surface (PDB-id 2PYO[52]) using 12 ambiguous interaction restraints (AIR) corresponding to mapped binding interface and one 1 unambiguous distance restraint corresponding to the observed crosslink. Active residues on the nucleosome were defined based on significant intensity and/or CSPs according to Fig. 2c. Active residues for RNF168$^{RING}$ were defined based on the mutagenesis experiments (R57 and R63). Passive residues were defined automatically by HADDOCK. Because of the small number of restraints for RNF168 the default random exclusion of 50% of the AIRs was switched off. The XL-MS restraint was defined as 28 Å Cα-Cα distance between RNF168 K46 and H2B K105. At each stage of the docking process all solutions were filtered based on their SASD for the crosslinked residues. Solution with SASD > 35 Å were rejected. Rejected solutions had on average 38.2 Å SASD, while accepted structures had SASD of 11.1 Å on average. Of the 200 water-refined solutions, 99.5% were clustered based on fraction of common contacts in two conformations (see Supplementary Fig. 4 for statistics).

To model the E2-E3-nucleosome complex, we first extracted 12 E2-E3 structures from the PDB database with either high structural similarity (PDB-id 4S30 and 4R8P) or high sequence similarity of the E3 to RNF168$^{RING}$ (PDB-id 2YHO, 3EB6, 3HCT, 3RPG, 4A49, 4AP4, 4QPL, 4TKP, 5FER, 5VNZ) using either the PDBeXplore or PDBeFOLD web-service. These 12 structures were superimposed and analyzed for conserved E2-E3 interactions (see Supplementary Fig. 5). The 13 conserved interactions were subsequently used as AIRs, specifying only the pairwise interactions at the residue-level, to dock the E2 UbcH5c (PDB-id 1X23) to RNF168. In addition, the linch-pin hydrogen bond between the RNF168 R55 side chain and E2-Q92 backbone was enforced. The E2 was docked to the ensemble of 10 best scoring solutions for the RNF168$^{RING}$-nucleosome complex, in which *Dm*. H2A N18 and *Dm*. H2B S121 were mutated in silico to match the *Hs*. H2A and H2B sequence. All solutions cluster in a single conformation of the E2-E3-nucleosome ternary complex. In all cases, the 20 best scoring solutions were analyzed for conserved intermolecular interactions.

**Molecular graphics**. All molecular graphics were prepared using PyMOL (The PyMOL Molecular Graphics System, Version 1.4, Schrödinger, LLC). Electrostatic surfaces were calculated using the adaptive Poisson-Boltzman solver[53] and the AMBER force field.

**Reporting summary**. Further information on research design is available in the Nature Research Reporting Summary linked to this article.

## Data availability

NMR data that support the findings of this study have been deposited in the BMRB with the accession codes 27547 for the backbone assignment of H2A in the H2A/H2B dimer, 27791 and 27792 for the titration using histone H2A- or H2B-labeled H2A/H2B dimers and 27786 for the titration using H2A- and H2B-labeled nucleosomes (http://www.brmrb.wisc.edu.gov). Proteomics cross-linking data are available via ProteomeXchange

under accession code PXD012723 (http://www.ebi.ac.uk/pride/archive). Structural models for RNF168-RING-nucleosome complex and the UbcH5c-RING-nucleosome complex are in the PDB-Dev database with the accession codes PDBDEV_00000028 and PDBDEV_00000029 (http://pdb-dev.wwpdb.org). All other data that support the findings of this study are available from the corresponding authors upon reasonable request. The source data underlying Figs. 1–4 and Supplementary Figs. 1, 2 and 4 are provided as a Source Data file. A reporting summary for this article is available as a Supplementary Information file.

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

## Acknowledgments

We thank Alma Svatos, Ivar Dilweg for their help in the assignments of H2A. We thank the uNMR-NL National Roadmap Large-Scale Facility of the Netherlands (NWO grant 184.032.207) for access to the 950 MHz NMR spectrometer and the Monash Biomedical Proteomics Facility of Monash University for access to mass-spectrometry equipment. This work was supported through a VIDI grant from The Netherlands Organization for Scientific Research (NWO) [723.013.010] to H.v.I. and NWO-TOP grant [714.012.001] and ERC advanced grant Ubiquitin balance to T.K.S.

## Author contributions

V.H. and M.U. designed and performed the experiments. H.Z., J.E., I.A., and U.B.P contributed to experiments and discussions. H.v.I C.D., and T.K.S. designed and supervised the experiments. H.v.I., V.H., and M.U. wrote the manuscript, with assistance from T.K.S. All authors critically read the manuscript.

## Additional information

**Competing interests:** The authors declare no competing interests.

