## [Peer Review File · Nature Communications]

Reviewers' comments:

Reviewer #1 (Remarks to the Author):

Summary

Site-specific monoubiquitylation of nucleosomal histones is an emerging topic with great interest in the field. In this manuscript, the authors use NMR spectroscopy and biochemical data to generate a structural model of the E3 ubiquitin ligase RNF168 RING domain (henceforth referred to as RNF168) bound to a nucleosome core particle (NCP). The main conclusion of the study is that an arginine-rich helix in RNF168 binds to the acidic patch of the NCP in a way that places the E2 active site directly over the substrate lysine (K15) of histone H2A. The model is interesting because it appears to explain the known lysine selectivity of RNF168 and because it reinforces what is becoming a common theme in RING E3 dependent nucleosome modification (Ring1b, BRCA1, Bre1); intricate electrostatic interactions between residues on the RING domain of the E3 and the nucleosome acidic patch govern the orientation of the E2 binding interface to direct substrate specificity.

The authors state that attempts to obtain a crystal or cryo-EM structure of the RNF168/NCP complex have failed, so a model based on NMR data and a small number of mutants was generated. These are difficult NMR experiments and the data presented are of high quality. The experimentally-based modeling using the HADDOCK protocol did not find a unique solution, but rather three clusters of possible models with similar energies. To distinguish among the clusters, the authors docked an E2 using the canonical RING surface (based on other experimentally-determined RING/E2 structures). This is a sensible approach, but it should be remembered that what results is a model based on models and care should be taken not to push interpretations too far (see below). The final model places the E2 active site closest to one of the known Lys residues that RNF168 modifies. Again, this is a sensible approach, but the model was chosen based on its being close to a relevant K15, so it seems circular to conclude that the model explains the selectivity towards K15.

The authors test their structural model by mutating Arg residues in the RNF168 helix and show that their mutation leads to a loss of ubiquitylation by RNF168. The results support the main conclusion that this helix, an element not found in other RING structures, plays a central role in NCP recognition by RNF168, but do not speak to substrate selectivity. For example, do any of the mutant RNF168 species modify a different lysine (on another histone?). In pursuit of the selectivity question, the authors identify a key mutation on histone H2B that distinguishes RNF168 binding to the NCP from that of Ring1b – another E3 that binds to the acidic patch and ubiquitylates H2A K119 on the other side of the nucleosome from H2A K15. The difference in orientation of the RING of RNF169 and the dimeric RING1b/Bmi1 RINGs is proposed to arise from differences in the electrostatics of the surfaces of the two RING E3s. While the electrostatic surfaces presented in a supplemental figure do support the differences, the explanation is somewhat unsatisfying. Presumably, there are favorable or unfavorable interactions at the edges of the electrostatic areas that help to give rise to the different orientations of the two RINGs—that is, what determines that the basic helix binds in the orientation observed rather than in the opposite one?

Overall, the paper presents high quality data on a challenging system that was refractory to crystallography, for which the authors are to be commended. There are, however, some additional ways that the authors could verify their model that would strengthen their conclusions, discussed below. With those improvements in hand, the paper will add important new information to the small but growing body of knowledge pertaining to E3-ligase mediated chromatin regulation. As such, it will be of interest to a wide audience.

Specific Comments/Suggestions

1. Figure 1C shows a histogram of intensity loss in the ¹⁵N-HSQC spectrum of H2A/H2B dimer at

0.65 molar ratio of RNF168. At this point, there is substantial intensity loss across the entire spectrum. The additional contribution to linewidth (and therefore intensity loss) of peaks that experience chemical shifts due to binding (over the general broadening due to complex formation) makes intensity measurements at smaller ratios of binding partner a better way to identify resonances in a binding site with more confidence. These should be shown in place of, or in addition to, the data presented.

2. ¹³C-methyl spectra are less prone to general broadening than ¹⁵N spectra and could give additional resolution of the binding site if ¹³C-H2A/H2B dimers were used. Presumably, resonance assignments for the methyl peaks are already available for the dimer. If not, I don't think it's worth the trouble to assign them, but more resolution in the methyl spectrum could be helpful.

3. While the authors provide mutagenesis data to implicate residues on the basic helix of RNF168 in NCP binding (R63 and R57), they do not directly show that these residues are necessary for NCP binding. Additional structural and/or biochemical data would strengthen this claim. One experiment that may accomplish this is an NMR titration of RNF168 harboring basic helix mutations. Ideally, these mutants would not produce specific chemical shift perturbations in the H2A/H2B dimer or NCP acidic patch. Alternatively (or additionally), a decrease in NCP binding in a gel-shift assay with RNF168 basic helix mutants would also strengthen the case for a critical role in NCP binding. Finally, as mentioned above, NCP Ub assays using mutant RNF168 and probing for other histone modifications could be insightful.

4. The authors state that the E2 UbcH5c does not make interactions with the NCP based on their model but there is no experimental test of this prediction. As the E2 plays an active role in the case of Ring1b-dependent NCP ubiquitylation, this is a point worth testing, at least using the E2 mutants used in the RING1b study in NCP Ub assays.

5. In the discussion section, the authors discuss specific side-chain interactions (E3-Ub) and rotamers (R55 of RNF168) observed in their model as well as the possibility of an active role of the substrate in facilitating ubiquitin transfer. Given the sparse data on which the model was generated, and the nature of the model itself, we advise caution here. For example, the authors state that the side-chain of H2A K15 is making a hydrogen bond with the E2 "gatekeeper" residue D117 promoting its deprotonation and making the E2~Ub conjugate more reactive. The notion of substrate-dependent activation of ubiquitin transfer, while intriguing as an idea, is not yet well supported in this study or elsewhere in the Ub field. Without a straightforward way to test the hypothesis, this discussion and associated figure(s) seems like a reach too far.

Minor Revisions

In the course of evaluating this manuscript we found several discrepancies in residue numberings and figure references in text. These should be carefully reviewed and corrected.

Reviewer #2 (Remarks to the Author):

In this study Horn et al. use NMR spectroscopy, molecular modeling, mutagenesis, and ubiquitination assays to investigate the structural basis underlying the known specificity of RNF168 for H2AK13/15. While this is an important question, a very nice approach, and some of the results are quite interesting, overall enthusiasm was substantially dampened by the rationale for the final structural model. Ultimately, it does not seem that the structural basis for specificity has been resolved here. Major and minor concerns are summarized below.

A major concern regards the analysis of the HADDOCK results. Here, three clusters are obtained with opposite positioning of the RING and thus E2, that must be resolved. While clusters 2 and 3 would position the E2 towards K13/15, cluster 1 would position the E2 towards the complete opposite side of the nucleosome near K119. Notably, cluster 1 is the best scoring cluster according to the HADDOCK score and Z-score. Though cluster 2 does not have a substantially different score,

it is still less favorable than cluster 1. Cluster 3, which is chosen for all further analysis actually has the worst HADDOCK score of all three clusters and a positive Z-score. Thus, it does not seem justified to ignore the results of cluster 1 and choose cluster 3 for analysis. Though RNF168 is indeed known to be specific for K13/15, this study is supposed to reveal the molecular mechanism for this specificity. Thus, using the specificity to justify the choice of structural model without any further evidence does not seem valid. Though the H2A E113A mutation is very cool and does indeed indicate a distinguishing molecular factor as compared to PRC1 RING1B activity, it does not seem that this can help in ruling out cluster 1 either. This must be resolved before these models can be understood with respect to specificity, as it could be that another factor not included in this model may be determining which orientation is preferred, which would actually be the determining factor of specificity.

Another major point regards how various mutations may be altering specificity. While the assays used nicely show the level of ubiquitination being achieved they do not address if specificity is changing upon mutation. Again, for determining the mechanism of specificity this is important.

Minor points include the following:

The structural stability of mutant constructs throughout the manuscript needs to be validated. Though these are in charged surface residues some evidence that these are properly folded should be provided. Additionally, binding assays with these mutants would be very informative to determine if it is nucleosome binding or Ub transfer that is impaired.

The switch between Dm and Hs numbering of the histones is confusing and seems unnecessary.

Figure 1C and 1E are very difficult to compare. While it would take up more room it would be nice to have separate plots for H2A and H2B such that it would be easy to compare the ILV results with the 15N results.

Though the labeling of Ub is clearly described as is the assay set up, the detection of assay results is not thoroughly laid out in the methods.

We thank the reviewers for their positive and constructive comments. To address their comments, we have added the following experiments to the manuscript:

- A crosslinking mass-spectrometry study to better define the RNF168 binding mode, resulting in a single, dominant cluster of solutions that explains the specificity for H2A K15 (new Figure 2f, Figure 3 and Supplementary Figures S3 and S4).
- Additional NMR titration experiments of RNF168 to H2A/H2B dimers with H2B labeled to more completely map the binding interface (new Figure 1b, c and Figure S1)
- A set of CPMG relaxation dispersion experiments to better discriminate specific from non-specific binding to histone H2A/H2B dimers (new Figure 1e and Figure S1).
- 1D NMR experiments of the wild-type, R63A and R67A/R68A RNF168 mutants to verify their folding (new Figure S2).
- Gel-shift binding assays of the wild-type, R63A and R67A/R68A RNF168 mutants to show the impact of the mutation on nucleosome binding (new Figure S2).
- Additional NMR titration of UbCH5c to H2A/H2B dimers with H2B labeled, in presence of RNF168, in an attempt to map the E2 binding surface on the histones (new Figure S7)

Additional new experiments are shown below in the point by point response to reviewers. A version of the manuscript with textual changes highlighted has been added.

Reviewer 1

To distinguish among the clusters, the authors docked an E2 using the canonical RING surface (based on other experimentally-determined RING/E2 structures). This is a sensible approach, but it should be remembered that what results is a model based on models and care should be taken not to push interpretations too far (see below).

We have taken this comment at heart and moved the more tentative aspects our discussion of the E2-E3-nucleosome model out of the main text into a Supplementary Figure and rephrased the text (new Figure S6).

The final model places the E2 active site closest to one of the known Lys residues that RNF168 modifies. Again, this is a sensible approach, but the model was chosen based on its being close to a relevant K15, so it seems circular to conclude that the model explains the selectivity towards K15.

We understand this concern. We therefore used crosslinking mass-spectrometry to identify intermolecular crosslinks that could help to resolve this issue. From three replicate experiments one reproducible interlink was found, between RNF168 K46, flanking the Arg-rich helix, and H2B K105, which lines the acidic patch. Combining this crosslink with the NMR and mutagenesis data in the docking results in a single dominant cluster of solutions, containing 187 out of 200 solutions. Superposition of this integrative structure with known E3-E2-Ub structures and docking with UbCH5c show that the E3 positions the E2 active site directly on top of the target lysine. The new data and extensive analysis are presented in Figures 2F (schematic overview of all obtained crosslinks); Figure S3 (analysis and quality control of the observed crosslinks); Figure 3 (the refined structural model); Figure S4 (docking statistics and validation of the crosslink). The main text has been adapted accordingly.

The authors test their structural model by mutating Arg residues in the RNF168 helix and show that their mutation leads to a loss of ubiquitination by RNF168. The results support the main conclusion that this helix, an element not found in other RING structures, plays a central role in NCP recognition by RNF168, but do not speak to substrate selectivity. For example, do any of the mutant RNF168 species modify a different lysine (on another histone?).

We apologize for not describing the ubiquitination assay clearly. In this assay, TAMRA-labeled ubiquitin is used and the bands on the gel are imaged using the TAMRA fluorescence. Thus, if ubiquitination were to shift from H2A to another H2A residue or to H2B for instance, still a prominent band would show, at the position of H2A-Ub or H2B-Ub. For the R63A and the R67A/R68A mutant band intensity for ubiquitinated histones is negligible to very low, showing that no lysine is efficiently ubiquitinated. We can thus rule out that specificity rather than activity has changed. We have clarified the description of these experiments in main text and Figure legends.

The difference in orientation of the RING of RNF169 and the dimeric RING1b/Bmi1 RINGs is proposed to arise from differences in the electrostatics of the surfaces of the two RING E3s. While the electrostatic surfaces presented in a supplemental figure do support the differences, the explanation is somewhat unsatisfying. Presumably, there are favorable or unfavorable interactions at the edges of the electrostatic areas that help to give rise to the different orientations of the two RINGs—that is, what determines that the basic helix binds in the orientation observed rather than in the opposite one?

This is an excellent point, which we did not address clearly. Indeed, the changes in the shape of the basic surface may explain how each E3 fits best to the acidic patch. RNF168 features an elongated basic surface that fits well to the H2A α 2/H2B α C cleft and fills it entirely, with unique favorable interactions at the far end of the

acidic patch. RING1B has a more square shaped basic surface that uniquely allows interaction between R81 and H2A D72. In addition, a Glu at the equivalent position in RNF168 is incompatible with this orientation. Rather, it is placed close to a Lys in the opposite orientation, thus favoring this position. We have reworked the text and Figures to make this more clear and moved this part from the Result section to the Discussion, where it is more appropriate.

Specific Comments/Suggestions

1. Figure 1C shows a histogram of intensity loss in the ^{15}N -HSQC spectrum of H2A/H2B dimer at 0.65 molar ratio of RNF168. At this point, there is substantial intensity loss across the entire spectrum. The additional contribution to linewidth (and therefore intensity loss) of peaks that experience chemical shifts due to binding (over the general broadening due to complex formation) makes intensity measurements at smaller ratios of binding partner a better way to identify resonances in a binding site with more confidence. These should be shown in place of, or in addition to, the data presented.

We have added the intensity ratios for the 25% added spectrum to the plot in Figure 1c. We would like to emphasize that the signal loss in this case is mainly due to precipitation of the complex. To gain more confidence in the mapped binding site we also titrated RNF168 to H2A/H2B dimers with H2B labeled. Because this sample was perdeuterated, the data are of very high quality and in fact allowed to discriminate specific and non-specific binding modes based on CPMG relaxation dispersion data.

2. ^{13}C -methyl spectra are less prone to general broadening than ^{15}N spectra and could give additional resolution of the binding site if ^{13}C -H2A/H2B dimers were used. Presumably, resonance assignments for the methyl peaks are already available for the dimer. If not, I don't think it's worth the trouble to assign them, but more resolution in the methyl spectrum could be helpful.

As mentioned under point 1, the main issue with the dimer titration is the lack of stability of the complex resulting in precipitation of complex. This would equally affect ^{13}C -methyl spectra (for which we don't have the assignments in the dimer context). In a way, the requested experiment has already been done, only in the context of the nucleosome, using ILV-labeled H2A/H2B (now presented in Figure 2a). In the nucleosome setup, the RNF168 complex does not precipitate and up to 2 equivalents of RNF168 could be added.

3. While the authors provide mutagenesis data to implicate residues on the basic helix of RNF168 in NCP binding (R63 and R57), they do not directly show that these residues are necessary for NCP binding. Additional structural and/or biochemical data would strengthen this claim.

One experiment that may accomplish this is an NMR titration of RNF168 harboring basic helix mutations. Ideally, these mutants would not produce specific chemical shift perturbations in the H2A/H2B dimer or NCP acidic patch. Alternatively (or additionally), a decrease in NCP binding in a gel-shift assay with RNF168 basic helix mutants would also strengthen the case for a critical role in NCP binding.

Indeed, we failed to address this. For R57, previous work by Mattioli et al showed by gel-shift assay that this residue is critical for nucleosome binding, reproduced here below. We have now included gel-shift data for R63 as well indicating strongly reduced binding affinity (Figure S2).

Figure 2f from Mattioli et. al Nat. Comm. 5, p. 3291 (2014) showing gel-shift assay for R57D mutant RNF168-RING binding to nucleosomes.

Finally, as mentioned above, NCP Ub assays using mutant RNF168 and probing for other histone modifications could be insightful.

As explained above, the assay visualizes TAMRA fluorescence so is sensitive to any ubiquitinated histone.

4. The authors state that the E2 UbcH5c does not make interactions with the NCP based on their model but there is no experimental test of this prediction. As the E2 plays an active role in the case of Ring1b-dependent NCP ubiquitination, this is a point worth testing, at least using the E2 mutants used in the RING1b study in NCP Ub assays.

We tested this suggestion via an NMR titration experiment. As RNF168/UbcH5c is active on dimers, we checked by NMR whether UbcH5c can bind to H2A/H2B in the presence of sub-stoichiometric amounts of RNF168. We failed to identify a clear interaction of the E2 to histone surface. The few residues that showed an effect, were also affected by addition of RNF168, complicating the interpretation (see new Supplementary Figure 7). This leaves open the possibility that E2 binding is stabilized within the nucleosome context. Such may be compatible with our model for the E2-E3-nucleosome complex, indicating a contact to the DNA and few histone residues (see Figure 3f). Activity data in the 2014 Mattioli et al. Nat. Comm. paper showed increased activity on nucleosomes compared to dimers, which in hindsight is suggestive of E2-DNA contacts. Further detailed investigations on the E2-substrate interaction are, we feel, beyond the scope of this paper, where we have focused on defining the E3-substrate interaction. We have added this discussion to the text.

5. In the discussion section, the authors discuss specific side-chain interactions (E3-Ub) and rotamers (R55 of RNF168) observed in their model as well as the possibility of an active role of the substrate in facilitating ubiquitin transfer. Given the sparse data on which the model was generated, and the nature of the model itself, we advise caution here. For example, the authors state that the side-chain of H2A K15 is making a hydrogen bond with the E2 "gatekeeper" residue D117 promoting its deprotonation and making the E2-Ub conjugate more reactive. The notion of substrate-dependent activation of ubiquitin transfer, while intriguing as an idea, is not yet well supported in this study or elsewhere in the Ub field. Without a straightforward way to test the hypothesis, this discussion and associated figure(s) seems like a reach too far.

We moved the more tentative aspects our discussion of the E2-E3-nucleosome model out of the main text into a supplementary Figure and rephrased them (new Figure S6).

Minor Revisions

In the course of evaluating this manuscript we found several discrepancies in residue numberings and figure references in text. These should be carefully reviewed and corrected.

We carefully proofread the manuscript and updated all figure references.

Reviewer #2 (Remarks to the Author):

In this study Horn et al. use NMR spectroscopy, molecular modeling, mutagenesis, and ubiquitination assays to investigate the structural basis underlying the known specificity of RNF168 for H2AK13/15. While this is an important question, a very nice approach, and some of the results are quite interesting, overall enthusiasm was substantially dampened by the rationale for the final structural model. Ultimately, it does not seem that the structural basis for specificity has been resolved here. Major and minor concerns are summarized below.

A major concern regards the analysis of the HADDOCK results. Here, three clusters are obtained with opposite positioning of the RING and thus E2, that must be resolved. While clusters 2 and 3 would position the E2 towards K13/15, cluster 1 would position the E2 towards the complete opposite side of the nucleosome near K119. Notably, cluster 1 is the best scoring cluster according to the HADDOCK score and Z-score. Though cluster 2 does not have a substantially different score, it is still less favorable than cluster 1. Cluster 3, which is chosen for all further analysis actually has the worst HADDOCK score of all three clusters and a positive Z-score. Thus, it does not seem justified to ignore the results of cluster 1 and choose cluster 3 for analysis. Though RNF168 is indeed known to be specific for K13/15, this study is supposed to reveal the molecular mechanism for this specificity. Thus, using the specificity to justify the choice of structural model without any further evidence does not seem valid.

Though the H2A E113A mutation is very cool and does indeed indicate a distinguishing molecular factor as compared to PRC1 RING1B activity, it does not seem that this can help in ruling out cluster 1 either. This must be resolved before these models can be understood with respect to specificity, as it could be that another factor not included in this model may be determining which orientation is preferred, which would actually be the determining factor of specificity.

We understand this concern and therefore used crosslinking mass-spectrometry to identify intermolecular crosslinks that could help to resolve this issue. From three replicate experiments one reproducible interlink was found, between RNF168 K46, flanking the Arg-rich helix, and H2B K105, which lines the acidic patch. Combining this crosslink with the NMR and mutagenesis data in the docking results in a single dominant cluster of solutions, containing 187 out of 200 solutions. Superposition of this integrative structure with known E3-E2-Ub structures and docking with UbcH5c show that the E3 positions the E2 active site directly on top of the target lysine. The new data and extensive analysis are presented in Figures 2f (schematic overview of all obtained crosslinks); Figure S3 (analysis and quality control of the observed crosslinks); Figure 3 (the refined structural model); Figure S4 (docking statistics and validation of the crosslink). The main text has been adapted accordingly.

Another major point regards how various mutations may be altering specificity. While the assays used nicely

show the level of ubiquitination being achieved they do not address if specificity is changing upon mutation. Again, for determining the mechanism of specificity this is important.

We apologize for not describing the ubiquitination assay clearly. In this assay, TAMRA-labeled ubiquitin is used and the bands on the gel are imaged using the TAMRA fluorescence. Thus, if ubiquitination were to shift from H2A to another H2A residue or to H2B for instance, still a prominent band would show, at the position of H2A-Ub or H2B-Ub. For the R63A and the R67A/R68A mutant band intensity for ubiquitinated histones is negligible to very low, showing that no lysine is efficiently ubiquitinated. We can thus rule out that specificity rather than activity has changed. We have clarified the description of these experiments in main text and Figure legends.

Minor points include the following:

The structural stability of mutant constructs throughout the manuscript needs to be validated. Though these are in charged surface residues some evidence that these are properly folded should be provided.

We recorded 1D NMR spectra of wild-type, R63A, and R67A/R68A to verify their folding (Figure S2).

Additionally, binding assays with these mutants would be very informative to determine if it is nucleosome binding or Ub transfer that is impaired.

Indeed, we failed to address this. For R57, previous work by Mattioli et al. showed by gel-shift assay that this residue is critical for nucleosome binding, see the Figure in the response to reviewer 1. We have now included (Figure S2) gel-shift data for the R63A mutant RING domain, indicating strongly reduced binding affinity, and R67A/R68A mutant, indicating the binding mode is altered even though the apparent affinity is largely unchanged.

The switch between *Dm* and *Hs* numbering of the histones is confusing and seems unnecessary.

We now stick to *Dm*. numbering, with the exception of Figs. 3e, 3f and 5a, 5b where we label and show the target lysine as H2A K15 and discuss the E2-E3-nucleosome model based on human histones, including *Hs*. H2B K120. In these cases we explicitly include the *Hs*. designation. In the discussion of the H2B E110 mutant, we added reference to the equivalent *Hs*. residue.

Figure 1C and 1E are very difficult to compare. While it would take up more room it would be nice to have separate plots for H2A and H2B such that it would be easy to compare the ILV results with the 15N results.

In the revised manuscript, Figure 1 now presents all data concerning the dimer experiment, including a H2B NMR titration. The ILV data is now presented in Figure 2 which obviously does not help for comparison. We hope that the plot of Figure 2c in which all data is aggregated helps to compare the results.

Though the labeling of Ub is clearly described as is the assay set up, the detection of assay results is not thoroughly laid out in the methods.

This has been added.

REVIEWERS' COMMENTS:

Reviewer #1 (Remarks to the Author):

The revised version of the manuscript is greatly improved and the authors are to be commended for their responsiveness to previous critiques. New results, especially the crosslinking data, provide necessary and compelling evidence regarding the orientation of the RING on the acidic patch. The use of CPMG relaxation experiments as a proxy for specific vs. non-specific binding is clever and helps to validate somewhat weak CSPs in the binding experiments in the context of H2A/H2B dimers. Also, most of the speculative language has been moved to the discussion, and the requested clarifications have been made within the text. The manuscript now presents an interesting and convincing story that will be of interest to readers across several fields and represents an important contribution to the small but growing number of structural models of E3 ligases bound to nucleosomes. There remain some minor aspects of the current version that could be improved to enhance reader comprehension and appreciation of that data, as outlined below.

- 1) In Figures 1&2, percentages are given in the upper corner of the graphs that show the NMR titrations. What do the percentages mean? Wouldn't molar equivalents be a better parameter to report? Regardless of what the authors decide to use here, it should be explained/defined in the figure legend.
- 2) In Figure S1 the authors report CSP in Hz as opposed to PPM. While not technically wrong, this can be misleading to the average reader, especially as similar data in the main text is reported in PPM.
- 3) Figure 1 is quite cluttered. The image showing the lack of CSPs on the backside of the H2A/H2B dimer would benefit from being larger. As very few readers can appreciate or assess the CPMG data, these could be moved to the supplemental section.
- 4) Please expand on the rationale behind the CPMG experiments in a manner accessible to a non-NMR expert. This is a creative and somewhat non-standard approach that may be lost on most readers.
- 5) Please clarify what is meant by "10% trimmed mean" in figure legends. Is it 10% of the biggest outliers, or 10% of the points on either end of the data (20% total)? A quick internet search indicates that the 10% trimmed mean can indicate either of these things.
- 6) In our opinion, the NMR experiment looking for binding with UbCH5c is inconclusive and was unlikely to work given its level of difficulty. Although not completely necessary, an NCP ubiquitylation assay using a mutant such as UbCH5c K125/R128E could strengthen the validity of the final model.

R. E. Klevit

Reviewer #2 (Remarks to the Author):

(none)

We thank the reviewers for their positive comments and feedback. A point by point response is given below. A version of the manuscript with changes tracked has been added.

Reviewer 1

- 1) In Figures 1&2, percentages are given in the upper corner of the graphs that show the NMR titrations. What do the percentages mean? Wouldn't molar equivalents be a better parameter to report? Regardless of what the authors decide to use here, it should be explained/defined in the figure legend.

We have changed the annotation in Figure 1 to molar equivalents as suggested and explicitly defined in the figure legend. We adapted Figure S1 similarly.

- 2) In Figure S1 the authors report CSP in Hz as opposed to PPM. While not technically wrong, this can be misleading to the average reader, especially as similar data in the main text is reported in PPM

We have changed Figure S1 and S7 to report the CSPs in ppm's as suggested and adapted the legend accordingly.

- 3) Figure 1 is quite cluttered. The image showing the lack of CSPs on the backside of the H2A/H2B dimer would benefit from being larger. As very few readers can appreciate or assess the CPMG data, these could be moved to the supplemental section.

We have reordered and edited the panels in Figure 1 to reduce clutter, present the data in a more logical order and edited the panels to make more room for the image as requested. We adapted the legend accordingly. We kept the CPMG data in this panel to highlight this approach to discriminate specific and non-specific binding, also in light of the next point.

- 4) Please expand on the rationale behind the CPMG experiments in a manner accessible to a non-NMR expert. This is a creative and somewhat non-standard approach that may be lost on most readers.

We have rephrased the original explanation:

"Reasoning that specific binding should result in larger chemical shift differences and slower dissociation rates than unspecific binding, we used CPMG relaxation dispersion experiments to probe these parameters via their effect on the ^{15}N transverse relaxation rate, $R_{2,\text{eff}}$."

to:

"Suspecting that the effects on the latter surfaces are due to unspecific binding of RNF168^{RING}, we sought to experimentally verify this. Since specific binding should result in larger chemical shift differences and slower dissociation rates than unspecific binding, we reasoned that these binding modes would result in an appreciably different response in NMR CPMG relaxation dispersion experiments. In these experiments the ^{15}N transverse relaxation rate, $R_{2,\text{eff}}$, is measured in a way that is very sensitive to dynamic chemical shift changes and larger chemical shift differences will cause larger dispersion of relaxation values."

- 5) Please clarify what is meant by "10% trimmed mean" in figure legends. Is it 10% of the biggest outliers, or 10% of the points on either end of the data (20% total)? A quick internet search indicates that the 10% trimmed mean can indicate either of these things.

Indeed, this should have been specified. For peak intensity and CSP analysis, the values are trimmed from one end (i.e. leaving out the 10% smallest intensity ratios and the 10% biggest CSPs). This has been added to the Figure legends.

- 6) In our opinion, the NMR experiment looking for binding with UbCH5c is inconclusive and was unlikely to work given its level of difficulty. Although not completely necessary, an NCP ubiquitylation assay using a mutant such as UbCH5c K125/R128E could strengthen the validity of the final model.

It is indeed very hard to capture the E2-E3-dimer complex by NMR, we have rephrased the text to emphasize this. We thank the reviewer for the interesting suggestion and will keep it in mind for a future study.